

# Best Practices and Uncertainties in CH₄ Emission Quantification: Employing Mobile Measurements and Gaussian Plume Modelling at a Biogas Plant

Julia B. Wietzel[1], Piotr Korben[1], Antje Hoheisel[1] and Martina Schmidt[1]

[1] Institute of Environmental Physics, Heidelberg University, Heidelberg, Germany

*Correspondence to*: Julia B. Wietzel (jwietzel@iup.uni-heidelberg.de)

**Abstract.** The increasing number of biogas plants in Germany and Europe necessitate an appropriate strategy to quantify potential methane ($CH_4$) losses to ensure the sustainability of renewable energy production. In addition to high uncertainties

in emission factors, there is little information on the temporal variation in $CH_4$ emissions from biogas plants. In this study, long-term measurements of $CH_4$ emission rates at a biogas plant in Heidelberg, Germany, were performed over a period of eight years using mobile measurements combined with a Gaussian plume model (GPM). To increase the accuracy of the emission rate calculations and harmonize the dataset, the methodology was evaluated through six controlled methane release experiments. The results of these experiments demonstrated that our method exhibits an uncertainty lower than 30%, provided

that the following recommendations are followed: wind and solar radiation should be measured onsite, at least 10 transects should be driven at low speeds, and a minimum distance of 20 m should be maintained from the emission source. By integrating these improvements into long-term monitoring practices at a biogas plant in Heidelberg, we present a comprehensive and consistent dataset of mobile measurements from 26 campaigns. The data revealed only low temporal variations in $CH_4$ emission rates, which is probably due to the continuous operation of the biogas plant with stable procedures. Notably, the

average $CH_4$ emission rate was $5.9 \pm 0.5$ kg $CH_4$ h⁻¹. The successful integration of data acquired over eight years through multiple measurement setups increases the reliability of the dataset, providing valuable insights into methane emissions from biogas plants.

## 1 Introduction

Reducing greenhouse gas (GHG) emissions from anthropogenic sources is an important step in mitigating climate change. The

reduction in short-lived greenhouse gases such as methane ($CH_4$), which exhibit a greater global warming potential than that of carbon dioxide ($CO_2$), constitutes a particularly effective mitigation strategy because their reduction yields a rapid impact. The energy (34%) and waste and agriculture (59%) sectors jointly accounted for 93% of the global anthropogenic $CH_4$ emissions from 2010–2019 (Saunois et al., 2024). Anaerobic digestion of biodegradable material is one form of waste management. This process generates biogas comprising 50–70% $CH_4$, 30–50% $CO_2$ and small amounts of $H_2S$ and $NH_3$

(UNFCCC, 2017). As a renewable energy source, the production of biogas addresses two challenges: the need to manage and recycle increasing amounts of organic waste and the need to reduce the use of fossil fuels. However, notably, methane emissions from biogas plants can reduce the mitigation effect in the case of major leakages. It is therefore important to accurately estimate methane emission rates and analyse the conditions at facilities to determine mitigation benefits. While the number of biogas plants in Germany increased by only 20% between 2012 (8,300) and 2022 (9,900), the average plant capacity

almost doubled from 400 to 760 kWel (Fachverband Biogas, 2023). Several studies conducted in Europe (Adams et al., 2015; Baldé et al., 2016; Fredenslund et al., 2018; Scheutz and Fredenslund, 2019; Bakkaloglu et al., 2021; Wechselberger et al., 2023; Fredenslund et al., 2023; Wechselberger et al., 2025) have revealed that $CH_4$ losses (the percentage of the amount of $CH_4$ emitted to the amount of $CH_4$ produced) varying between 0.02 and 40.6% significantly affect the GHG balance at biogas plants. The large number of biogas production facilities in Germany and the wide range of methane losses reported in other



European studies highlight the need to monitor and quantify methane emissions. Measurements conducted at regular time intervals could provide a representative analysis of emissions at the measurement sites and their evolution over time. This has been demonstrated by Brilli et al. (2024) and Kumar et al. (2024) for landfills and by Johnson and Heltzel (2021) and IJzermans et al. (2024) for oil and gas facilities. Maldaner et al. (2018) determined methane emission rates from a digestate storage tank at a dairy manure biogas facility over one year using a micrometeorological mass balance approach. Long-term monitoring is

particularly useful for complex facility structures such as biogas plants, where $CH_4$ emissions are not expected to remain constant over time (Flesch et al., 2011; Hrad et al., 2015; Balde et al., 2016; Reinelt and Liebetrau, 2020).

Different measurement methods can be used to quantify the $CH_4$ emissions of individual sources, such as the tracer gas dispersion method (Scheutz and Fredenslund, 2019; Delre et al., 2018; Hrad et al., 2022), chamber measurements (Liebtrau et al., 2013), the mobile flux plane (Rella et al., 2015), inverse dispersion modelling (Hrad et al., 2022; Wechselberger et al.,

2025) and the Gaussian plume/OTM 33a method (Ars et al., 2017; Korben et al., 2022).

Analysing measurement data from mobile platforms via a dispersion model to calculate $CH_4$ emission rates provides the advantage that measurements can be conducted on roads outside installations and without access (Kumar et al., 2021, Bakkaloglu et al., 2021). The method used in this study is based on accurate $CH_4$ concentration measurements from a car or bicycle with a high temporal resolution combined with a Gaussian plume model (GPM). Although this method is relatively

cost-effective and allows a high sampling efficiency at individual sources, the model is subject to several sources of uncertainty, such as the choice of stability class, driving strategy and averaging method (Ars et al., 2017; Caulton et al., 2018; Kumar et al., 2021; Riddick et al., 2022). The GPM has great potential because of its easy and fast parameterization and application to the emissions of different sources measured during a mobile campaign. However, carefully testing the capabilities and limitations of this method and evaluating its uncertainties through controlled release experiments are

important. Such experiments have already been conducted in other studies to support the development, testing and improvement in atmospheric measurement and modelling techniques to determine, locate and quantify $CH_4$ emissions (Ars et al., 2017; Caulton et al., 2018; Kumar et al., 2021; and Morales et al., 2022). However, differences in local conditions and different measurement equipment require a detailed examination of the method to ensure that it is adapted to individual requirements.

This study presents several controlled methane release experiments conducted on six different days in 2018, 2020, and 2023 to assess and improve the accuracy and uncertainty in $CH_4$ emission rate determinations using a Gaussian plume dispersion model. We investigated the impacts of varying meteorological conditions, different measurement instruments and sampling strategies, and we provided a detailed evaluation. These experiments capture a range of experimental variations, which enables the analysis of their effects on emission rates. They also contributed to the compilation of a comparable dataset for our long-

term field campaigns at a biogas plant in Heidelberg, Germany. In the second part of the study, we present the time series of $CH_4$ emission rates from 26 measurement days over eight years at the same local biogas plant.

## 2 Site description and methods

### 2.1 Biogas plant

The investigated biogas plant is located in Heidelberg, in southwestern Germany. Figure 1 shows a map of the biogas plant

site, with the emission source location indicated as a blue dot. A zoomed-in schematic highlights the positions of the digesters, biogas storage unit, and combined heat and power (CHP) unit, while roads accessible by vehicles are marked in yellow. The plant, which was built in 2001, consists of two 500-$m^3$ anaerobic digesters, in which organic waste and maize silage (and further substrates) are converted into biogas under anaerobic conditions. The biogas plant specializes in the disposal of organic waste, such as that from the food industry and household waste. Approximately 1,770 t of biogas is produced annually and



converted into electricity and heat in a 500-kW CHP unit. A biogas storage capacity of approximately 1,500 m$^3$ ensures that the CHP unit operates continuously and that there are no major fluctuations in electricity production.

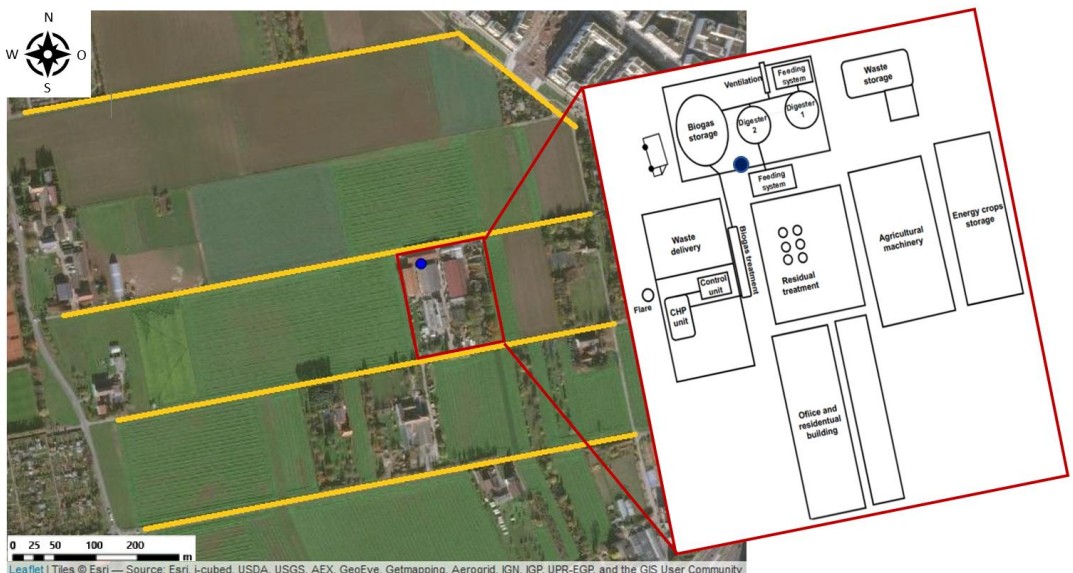

**Figure 1. Map of the biogas plant site near Heidelberg, with the emission source location selected for implementing the GPM marked**
**as a blue dot. A zoomed-in schematic of the plant shows the locations of the digesters, biogas storage, and CHP unit. Roads accessible for driving are marked in yellow. The map was created using Leaflet for R.**

Since August 2016, regular mobile measurements of CH$_4$ isotope ratios have been conducted at the biogas plant by direct sampling and with air core measurements in the plume, resulting in an average $\delta^{13}$CH$_4$ source signature of -62.4 ± 1.2‰
(Hoheisel et al., 2019). Since 2018, these mobile CH$_4$ measurements have focused on determining the CH$_4$ emission rate using a dispersion model. By reanalysing previous mobile measurements performed by Hoheisel et al. (2019), it is possible to determine the CH$_4$ emission rates for the entire period from August 2016 to August 2024 using a harmonized method. The CH$_4$ emission rates at this biogas plant were determined on a total of 26 measurement days over the eight-year period. These measurements were conducted at different distances, under different metrological conditions and with a variable measurement
setup.

### 2.2 Controlled release experiments

Between 2018 and 2023, CH$_4$ controlled release experiments were performed over six days in Mannheim and Heidelberg, Germany. The aim of these experiments was to analyse the accuracy and uncertainty in the CH$_4$ emission rates calculated with a Gaussian plume model in combination with mobile CH$_4$ measurements. The influences of the driving strategy, measurement
setup and meteorological parameters on CH$_4$ emission rate estimation were evaluated.

Three controlled CH$_4$ release experiments were conducted in November 2018, September 2020, and October 2020 in a parking lot in Mannheim (longitude: 49.470816°; latitude: 8.514822°; altitude a.s.l.: 89 m) (Fig. S1). The parking lot encompasses flat terrain with a size of 750 m × 250 m and no major obstacles. A local airport occurs to the north, and to the south, between the car park and the highway, there are trees and bushes.

Between October 11 and 13, 2023, a three-day controlled release campaign was conducted at a former airfield in Heidelberg, Germany (longitude: 49.391954°; latitude: 8.654846°; altitude a.s.l.: 105 m) (Fig. S2). The airfield is located southwest of



Heidelberg and offers a large open area (400 m × 400 m). As shown in Fig. S2, several hangars and a tower occur in the west, whereas an open area extends to the north and east. The site is surrounded by fields and individual houses. Two biogas plants are located 700 m northeast. To avoid possible influences of $CH_4$ emissions in the northeast, $CH_4$ release experiments were

performed at this site only under southerly or southwesterly winds.

The experiments involved the controlled release of $CH_4$ at rates ranging from 0.1 to 0.7 kg $CH_4$ h$^{-1}$ from a 10-L high-pressure cylinder containing pure methane (99.5% $CH_4$) followed by mobile measurement of the $CH_4$ mole fraction in ambient air at various distances downwind from the release point. The cylinder used to simulate the methane point source was connected to a flow meter (Yokogawa Rotameter, model RAGL) and 3 m of tubing. The end of the tubing was installed 1 to 2.5 m above

ground. As an additional control of the release rate, the $CH_4$ cylinders were weighed before and after each gas release process (balance model: DS30K0.1, Kern & Sohn GmbH, Balingen, Germany; readout precision of 0.1 g). Before the start of each controlled release of $CH_4$, the background $CH_4$ mole fraction was determined by measurements close to the release point. More details on the meteorological conditions, instrumentation and number of releases are provided in the Supplementary Materials.

### 2.3 Mobile measurement setup

Between 2016 and 2020, a cavity ring-down absorption spectrometer (CRDS, G2201-i, Picarro, Inc., Santa Clara, CA) was installed in a van for mobile, in situ measurements of the atmospheric mol fractions of $CH_4$, $CO_2$ and $H_2O$ as well as the isotopic compositions of $\delta^{13}CH_4$ and $\delta^{13}CO_2$ with a temporal resolution of 0.27 Hz. The mobile setup and calibration procedure have been described in detail by Hoheisel et al. (2019). Ambient air was pumped at a flow rate of 0.16 l min$^{-1}$ through a 1/4" Teflon tube from the inlet of the roof system at approximately 2.7 m above ground to the trace gas analyser inside the van.

While driving, the position was tracked using a GPS (Navilock, USB 2.0 Receiver SiRFstarIV). The time delay caused by the dead volume of the air inlet tubing and instrument setup was determined and corrected by measuring a small $CO_2$ pulse (breath test).

Since 2020, mobile $CH_4$ and $CO_2$ measurements were performed with an optical feedback-cavity enhanced absorption spectroscopy (OF-CEAS) trace gas analyser (LI-7810, LI-COR, Lincoln, USA), with a relatively high temporal resolution of

1 Hz and a flow rate of 0.31 L min$^{-1}$. A detailed description of the measurement setup and calibration procedures for this analyser has been given by Korben et al. (2022) and Wietzel and Schmidt (2023). The route was tracked using a GPS (BasicAirData, Google Commerce Ltd.). As in the previous setup, the air inlet for the trace gas measurements is installed on the roof of the measurement vehicle, and delay times are corrected during data analysis. Meteorological parameters were recorded with a 2D anemometer (Gill Instruments, UK) mounted on the roof of the vehicle near the inlet or/and a 3D ultrasonic

anemometer (USA-1 Sensor, Metek Meteorologische Messtechnik GmbH, Elmshorn, Germany) or a stationary weather station (Vantage Pro2 Davis Instruments) located near the source. Table S1 in the Supplementary Materials provides an overview of the different measuring instruments and the times they were used together with their specifications. Figure S3 shows the roof system with the air inlet and mobile anemometer mounted on the van as well as the 3D anemometer on a tripod.

### 2.4 Gaussian plume dispersion model

In this study, we employed a Gaussian plume model (GPM) to describe the dispersion of gas (here, $CH_4$) in the atmospheric boundary layer emitted from a point source. Assuming stable meteorological conditions and a source emitting at a constant rate over the observation period, the GPM describes the relationship between the measured $CH_4$ concentration (C) downwind of the source as a function of the distance (x) and the emission rate (Q) (Turner, 1970; Hanna et al., 1982). Specifically, the GPM is based on the spatial distribution described by a combination of normal distributions in both the vertical and horizontal

planes of $CH_4$ downwind of the source and the associated meteorological measurements. Through the use of this approach, the emission rate from the source can be calculated via Equation (1).



$$C(x, y, z) = \frac{Q}{2\pi\sigma_y\sigma_z u} \exp\left[-\frac{y^2}{2\sigma_y^2}\right] \left\{\exp\left[-\frac{(z-h_s)^2}{2\sigma_z^2}\right] + \exp\left[-\frac{(z+h_s)^2}{2\sigma_z^2}\right]\right\} \tag{1}$$

This equation accounts for a reflective ground surface. Data on the wind speed (u; m s$^{-1}$), wind direction (WD; °), temperature (T; °C), and CH$_4$ mole fraction above the background level in the plume (C; μg m$^{-3}$) at distance (x), location and height of the

CH$_4$ source (h$_s$; m) and the Pasquill stability class (Pasquill, 1961) were used. Typically, the stability class can be assigned from extremely unstable to stable meteorological conditions in the boundary layer using the wind speed and a measure of the solar radiation (Turner 1970). The dispersion coefficients σ$_y$ and σ$_z$ along the horizontal (y) and vertical (z) directions can then be derived from the determined Pasquill–Gifford stability class (PGSC) (Table S2) and the downwind distance (x) from the source on the basis of the Briggs parameterization (Table S3) (Hanna et al., 1982; Griffiths, 1994). Since the GPM does not

provide the background mole fraction but only the CH$_4$ excess caused by a point source, the background mole fraction must be determined and subtracted from the series of mobile measurements. As the background can vary over time, it was calculated using a variable background fit (Ruckstuhl et al., 2001, Wietzel and Schmidt, 2023). In practical applications, transects are traversed with a measurement vehicle downwind passing through the plume perpendicular to the wind direction while the corresponding CH$_4$ mole fraction is measured. Individual recordings of the emission plume are then compared with the

theoretical approximation of the average CH$_4$ dispersion provided by the GPM. Since the output of the GPM is linear with respect to Q, we set the initial emission rate (Q) to 1 g CH$_4$ s$^{-1}$ to model the corresponding methane mole fraction (C). The ratio of the integral of the mole fraction above the background along a given transect and the integral of the peak modelled by the GPM corresponds to the estimated emission rate (Q) (Mønster et al., 2014; Korben et al., 2022). Each individual peak was examined to exclude cases with unfavourable trajectories, where part of the peak was cut off. A value of the determination

coefficient (R²) greater than 0.5 was used as the criterion for a valid comparison between the measured and modelled plumes for each transect to accept the transects (Korben et al., 2022). Only transects that passed the quality check were included in the calculation of the average emission rate for a specific source. A typical example of a time series with 10 transects and a background fit in blue, recorded with the OF-CEAS trace gas analyser, is shown in Fig. 2a. Figure 2b shows the modelled horizontal transects of the Gaussian plume of a source for three emission rates Q (1 g CH$_4$ s$^{-1}$ (blue), 0.5 g CH$_4$ s$^{-1}$ (green) and

0.1 g CH$_4$ s$^{-1}$ (orange)) at x = 100 m, z = 2.5 m, h$_s$ = 1 m, and u = 2.5 ms$^{-1}$ and stability class D. At this downwind distance of 100 m, the width of the plume is approximately 40 m. The width increases when σ$_y$ increases, which is the case at greater downwind distances and under more unstable wind conditions.

The GPM is a relatively simple model that assumes stable weather conditions and a well-defined plume originating from a point source. The neglect of surface turbulence and the parameterization of the PGSC can lead to significant uncertainties.

However, the advantage of the GPM is that many mobile measurements can be modelled in a relatively short time, the measurements do not depend on direct access to the site, and the model is relatively cost-effective and allows a high sampling efficiency at individual sources.



**(a)** 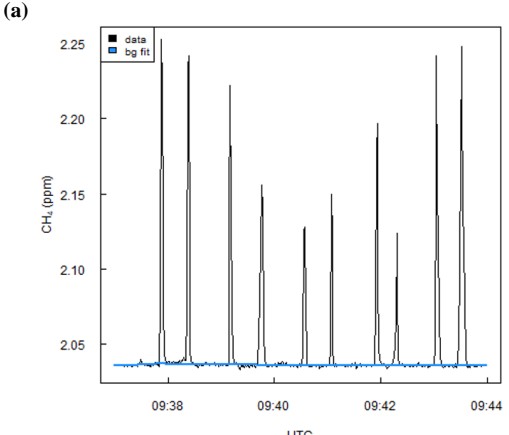  **(b)** 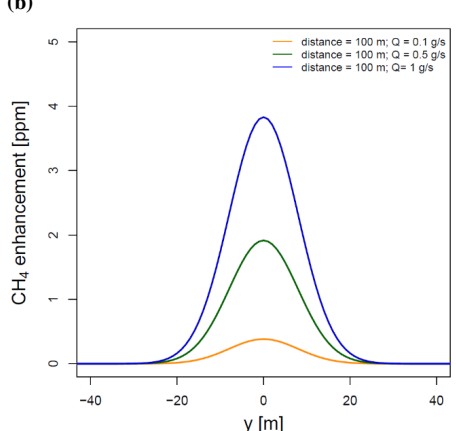

**Figure 2. (a) Measured CH$_4$ mole fraction (ppm) against time (hh:mm) during emission plume crossings recorded with the LI-7810**
**trace gas analyser. The calculated background mole fraction is highlighted in blue. (b) Model cross sections of the GPM during CH$_4$**
**plume crossing downwind of the source at a distance of x = 100 m with different input emission rates (blue, Q = 1 g CH$_4$ s$^{-1}$; green,**
**Q = 0.5 g CH$_4$ s$^{-1}$; yellow, Q = 0.1 g CH$_4$ s$^{-1}$).**

## 3 Results and discussion

The controlled CH$_4$ release experiments, which were performed on six days, were conducted under different meteorological
conditions (Table 1). This allowed the application of the Gaussian plume model for analysis under slightly stable and stable
atmospheric conditions. A total of 39 releases were conducted to test different parameters, develop the best strategy for
measurement and model evaluation, and determine the magnitude of uncertainties. The results without a major impact on the
emission determinations are only briefly mentioned in the Results section and then described in more detail with graphs in the
Supplementary Materials. The first subsection focuses on the driving strategy of the mobile measurements, CH$_4$ sampling
frequency and transect averaging, whereas the second subsection provides a description of the meteorological parameters and
their influence on the determined emission rates. In the last section, the results obtained are applied to calculate CH$_4$ emission
rates at the biogas plant near Heidelberg in Germany.

| ID | Date | Location | Instrument | Anemometer | Temperature [°C] | Meteorological conditions | Wind speed [m/s] | Number of release steps |
|-----|------------|------------|----------|--------------|-----|----------|-----|---|
| MA1 | 28.11.2018 | Mannheim | G2201-i | Metek | 5 | overcast | 2.8 | 5 |
| MA2 | 10.09.2020 | Mannheim | LI-7810 | Gill + Metek | 22 | sunny | 1.7 | 6 |
| MA3 | 22.10.2020 | Mannheim | LI-7810 | Gill + Metek | 19 | cloudy | 1.9 | 6 |
| HD1 | 11.10.2023 | Heidelberg | LI-7810 | Gill + Metek | 21 | sunny | 2.2 | 9 |
| HD2 | 12.10.2023 | Heidelberg | LI-7810 | Gill + Metek | 20 | overcast | 1.2 | 9 |
| HD3 | 13.10.2023 | Heidelberg | LI-7810 | Gill + Metek | 22 | sunny | 2.4 | 4 |

**Table 1. Overview of the controlled CH$_4$ release experiments performed between 2018 and 2023 in Mannheim (MA) and Heidelberg**
**(HD).**





### 3.1 Controlled release experiments: Analysis of the experimental setup, driving strategy and averaging methods

During the eight-year period of long-term measurements at the biogas plant, the mobile instrumentation setup was varied. Different wind sensors and two types of trace gas analysers were used (Section 2.3). Accurate measurement of atmospheric trace gas mole fractions, particularly the correct reproduction of the concentration peak during plume crossing, is essential for
the calculation of emission rates. Therefore, the response time of gas analysers is critical, especially in dynamic environments where trace gas mole fractions change rapidly. The CRDS analyser, which was used for mobile ambient air measurements of $CH_4$ and $CO_2$ from 2016 to 2020, provides a lower data acquisition frequency (0.27 Hz) and is more optimized for isotope measurements. In contrast, the OF-CEAS analyser, employed from 2020 onwards, provides a faster data acquisition time (1 Hz), which is better suited to applications that require rapid detection of concentration changes, such as flux measurements. It
is important to consider the response time of the instrument within the context of real-time measurements and how this might influence the accuracy of the emission rate calculations.

In our study, we investigated the response time during one mobile survey in which both the CRDS analyser and the OF-CEAS analyser were employed for simultaneous measurements with different temporal resolutions. The faster analyser (OF-CEAS) provided a higher mole fraction, whereas the peak measured with the slower analyser (CRDS) was broader (Fig. S4). If the
mole fraction in the ambient air changes rapidly, the instrument may miss the true peak height in real time. Takriti et al. (2021) also investigated the effect of the analyser response time on concentration measurements by conducting mobile surveys with two gas analysers. Similar to Takriti et al. (2021), we found that for our two analysers, the peak heights differed, but the area under the molar fraction curve integrated over time remained consistent. By using the peak area to calculate emission rates (Mønster et al., 2014; Korben et al., 2022), we can ensure that emission rates derived from measurements with our two types
of instruments remain consistent and comparable. This would make it possible to combine the time series of mobile measurements from both instruments to determine the emission rates at the biogas plant, covering a total period of eight years. This will be discussed again at the end of this section when all release tests are analysed in terms of changing instruments.

In addition, the influences of the driving speed and the choice of mobile vehicle (bicycle or car) were investigated in separate release experiments. No dependencies of the determined emission rate on these two modifications were found. A more detailed
description of this process can be found in the Supplementary Materials (Figs. S5+S6).

### Influence of the downwind distance on the emission rate estimates

As the locations of accessible roads are different for each $CH_4$ emission source, the plume is usually crossed at different distances. It is therefore important to analyse the estimated emission rate as a function of the distance between the emission
source location and the measurement point to account for this aspect during the planning phase. The Briggs parameterization (Briggs, 1973), which is used to calculate dispersion coefficients, is based on measurements for distances between 100 m and 10 km. Under stable meteorological conditions, distances of 100, 250 and 310 m between the release point and analyser inlet were considered, with a release rate of 0.6 kg $CH_4$ $h^{-1}$ and a wind speed > 3 m $s^{-1}$ (MA1). No significant difference between the emission estimates at the three distances was observed (Fig. S7). However, measurements closer to the emission source at
distances of 14 and 36 m were analysed during MA3, and a large range from 5 to 260 m was obtained during HD2. The data from MA3 already indicated that very close measurements at 14 m from the source could lead to overestimation of the emission rate by a factor of two (Fig. S8). In even more extreme cases, such as the scenario during HD2 at a distance of 5 m, it is no longer possible to realistically estimate the emission rate, which is overestimated by more than 1000% (Fig. S9).

This study shows that our model can only reproduce the actual release rate within an acceptable uncertainty range at a distance
of more than 20 m between the emission source and the measurement vehicle. This result differs from the studies of Day et al. (2014), who reported no dependence in four controlled release experiments at distances between 15 and 30 m from the source.





Rella et al. (2015) also reported no correlation between the estimated $CH_4$ emission rate and the wind direction at a distance of 15 to 150 m from oil and gas wells.

However, it should be noted that as the distance from the emission source increases, the model becomes less sensitive to
inaccuracies in the source location, which in turn affects the calculation of the modelled mole fraction and the estimated emission rate (Mønster et al., 2014). This is particularly important for field measurements, where the exact location of the emission source is not always known accurately. In this respect, with our measurement and modelling setup, we would only accept measurements obtained at a distance greater than 20 m from the emitter.

**Number of transects and data averaging**

By analysing the results of the controlled release experiments, an improved sampling strategy can be developed. During field campaigns, it is important to find a balance between minimizing the sampling time and maximizing the precision. On average, 60 s are needed to complete one transect (plume crossing). Like Caulton et al. (2018), we investigated the convergence of the determined $CH_4$ emission rate as well as the standard deviation as a function of the number of transects. For this purpose, data
from release experiment HD1, consisting of 30 transects, were used, and an increasing number of randomly selected transects were adopted from the set n times to calculate the corresponding average total emission rate. Figures 3a and 3b show box and whisker plots of the median and mean values, respectively. In some cases, significant deviations from the nominal release rates (dashed red line) were observed. Note that the uncertainties shown in the boxplot correspond to the distribution of the calculated mean or median values (uncertainty due to the limited number of transects) and therefore do not reflect the
measurement accuracy. However, by averaging over an increasing number of plume transects, the standard deviation is reduced, and the precision increases. A significant reduction in the variance in the results is observed for approximately 10 transects, after which a further increase in the number of transects only leads to a slow reduction in the variance in the results. The standard deviation decreases by 80% for 10 transects but only decreases by an additional 10% for 20 transects. On the basis of these observations from the field experiments, a minimum of 10 plume transects is recommended to decrease the
influence of atmospheric variability. A similar result has already been described by Caulton et al. (2018), who also recommended 10 transects.

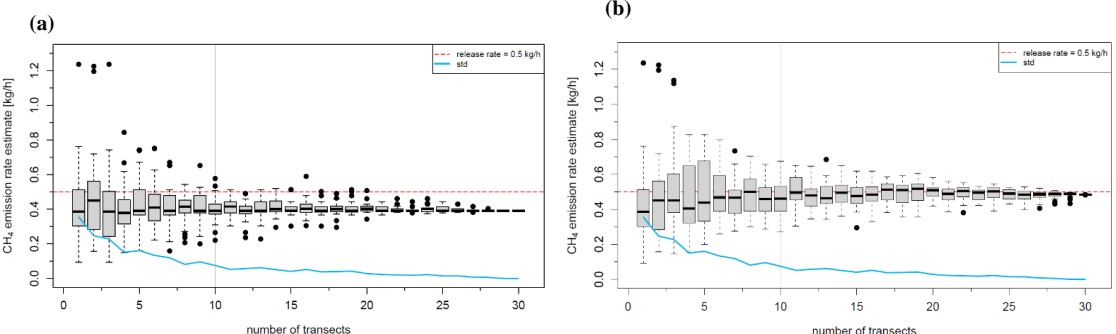

**Figure 3. Convergence of the (a) median and (b) mean plume rates by averaging randomly selected transects from one set conducted during release experiment HD1. The actual release rate is shown as a dashed red line. The box and whisker plots show the average,
25th and 75th percentiles, and minimum and maximum values of the means obtained for each number of transects. The outliers are shown as black dots. The standard deviation of the average values is shown in blue.**

When 10 or more transects are traversed, the question also arises as to whether calculating the mean or median is the appropriate averaging method to determine the emission rate per emission source. Figure 3 a and b clearly show that the mean is closer to
the actual release rate (dashed red lines). In this example, the median converges to a value of 22% below the $CH_4$ release rate,



whereas the mean is only 3% lower. Figures 4a and 4b show summaries of all the release experiments performed and evaluated between 2018 and 2023. As in the previous example, the mean provides a more accurate estimate than does the median, with an average difference of 35% compared with 43%. Note that the determined emission rates were not affected by the change in measurement setup between release experiments MA1 and MA2.


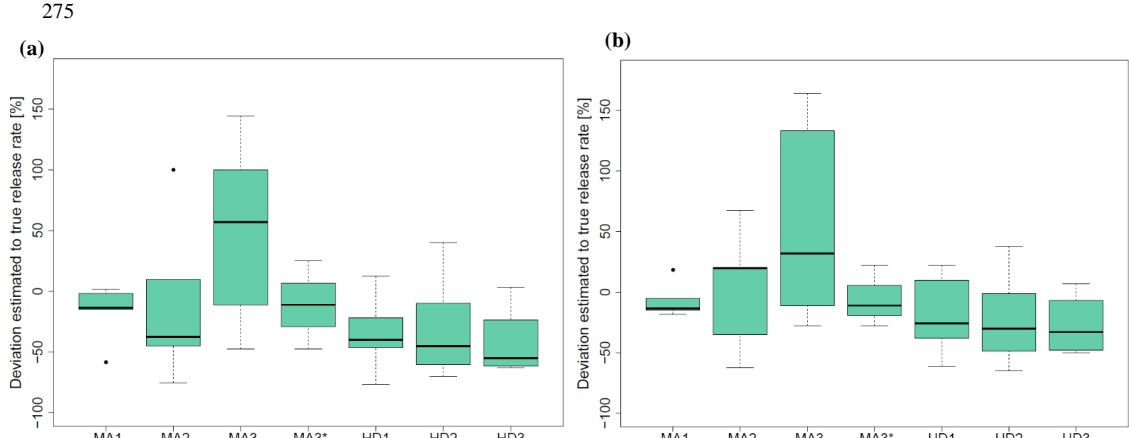

**Figure 4: Boxplot for the comparison of the determined methane emission rates during the different release experiments to the actual release rates for (a) median and (b) mean values. The boxplots show the first and third quartiles of the data, whereas the whiskers extend to the largest value that remains within 1.5 times the interquartile range. Release experiment MA3 is split into MA3, which includes all release processes, and MA3\*, which excludes release processes at a 14-m distance.**


### 3.1 Influence of meteorological parameters on the calculated emission rates

During the eight years of mobile surveys, different wind sensors, including stationary sensors (Davis and Metek) and mobile sensors (Gill), were used for the wind measurements. The influence of the use of these sensors on the emission rate results was analysed. For the application of the GPM, continuous in situ measurements of the wind speed with a sufficient temporal resolution and high accuracy are required as input parameters. The calculated emission rate is proportional to the wind speed and the stability class; therefore, the dispersion coefficients $\sigma_y$ and $\sigma_z$ are also dependent on the wind speed (Eq. (1)). In two comparison campaigns of wind sensors, all three sensors used over time were first set up as static sensors, and in a second comparison, a mobile sensor was compared with a static sensor during the HD3 release experiment.

In February 2019, simultaneous measurements were obtained using three weather stations located on the institute roof to compare the wind data (Kammerer, 2019). The Metek 3D anemometer was considered the most accurate and was employed as the reference instrument. Data were recorded in seconds (Metek and Gill) and minutes (Davis) and averaged to hourly mean data as wind vectors (Fig. S10). The results from the 2019 measurements indicated that the wind speeds measured by the Gill and Davis instruments were slightly lower than those recorded by the Metek instrument. However, these differences were not statistically significant. The differences in the wind direction were also not significant over the entire measurement period.

Figure 5 shows a comparison measurement in 2023 (HD3), which focuses on evaluating the performance of the Gill anemometer during vehicle motion, to verify the internal correction that accounts for the driving wind and vehicle alignment. Driving times are highlighted in grey. When both the Gill and the Metek anemometers were stationary, the data were comparable, with the Gill anemometer providing an average wind speed of $2.5 \pm 1.1 \text{ m s}^{-1}$ (wind direction: $174 \pm 26°$) and the Metek anemometer providing an average wind speed of $2.7 \pm 1.0 \text{ m s}^{-1}$ (wind direction: $169 \pm 20°$). The difference in the wind speed values was similar to that recorded in 2019. However, during vehicle movement (Fig. 5, highlighted in grey), especially during acceleration phases, the Gill instrument presented greater variability in terms of the wind speed and standard deviation,





with a value of 2.7 ± 1.7 m s$^{-1}$ and a wind direction of 207 ± 124°, than did the Metek instrument, with a value of 2.6 ± 1.1 m s$^{-1}$ and a wind direction of 176 ± 19°. Once the vehicle reached a constant speed, the wind measurements from the Gill instrument agreed with those from the stationary Metek instrument. The uncertainty in the Gill measurements (calculated as the standard deviation of the 1-s values) was significantly lower when the car did not move, especially for the wind direction. However, the averages were not affected.

Inaccurate wind data can introduce significant errors, as the Gaussian equation (Eq. (1)) shows a linear relationship between the inverse of the wind speed and the emission rate. This suggests that the relative uncertainty in the emission rate is likely similar to the relative uncertainty in the wind speed. Caulton et al. (2018) demonstrated that using carefully measured, onsite in situ wind data greatly enhances the accuracy of CH$_4$ emission rate estimates compared with relying on modelled wind fields, which may not accurately represent site or NOAA-provided wind data.

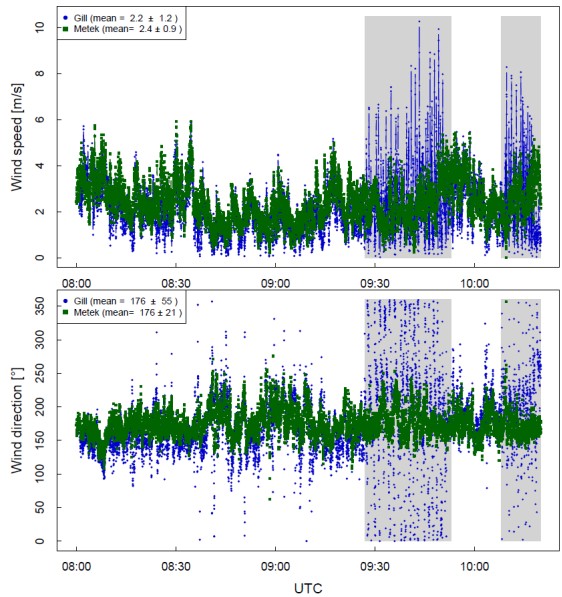

**Figure 5. Wind speed (top) and wind direction (bottom) measured with the mobile (Gill) and stationary (Metek) instruments during the HD3 release experiment (October 2023). The grey areas highlight the periods when the car was driving.**

Wind variability is a critical factor influencing the dispersion and detection of atmospheric plumes, often leading to deviations in the lateral positioning of transects. This phenomenon, referred to as the meandering effect, arises from atmospheric eddies that are larger than the diameter of the plume. Accurate determination of the wind direction is therefore essential for reliable emission rate calculations. Caulton et al. (2018), Kumar et al. (2021), and Korben et al. (2022) reported that the use of geographic coordinates to determine the wind direction from the location of the maximum peak concentration during a transect and the emitter can provide a more accurate representation of plume behaviour and a better reproduction of modelled peaks than the use of directly measured wind data can. We analysed data from ten transects during the HD1 release experiment using the two distinct anemometer setups: a stationary anemometer positioned at a fixed location (Metek) and a mobile anemometer mounted on a vehicle (Gill) moving along the transects. Figure 6a shows the anemometer data for the wind direction averaged across the transects as well as the modelled wind direction, which are based on geographical coordinates. The modelled wind direction and the stationary measurements (Metek) suitably agreed, whereas the data from the mobile wind sensor (Gill) deviated significantly. These differences are directly reflected in the calculated emission rates, as shown in Fig. 6b. The emission rates calculated using the data from the stationary anemometer (Metek) did not significantly vary when the measured

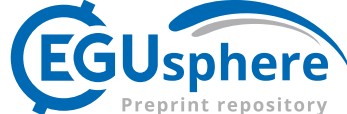



and modelled wind directions were compared, whereas the use of the measured mobile wind directions (orange bars) led to

significant overestimation of the emission rates. The approach proposed by Kumar et al. (2021) and adopted by Korben et al. (2022), which involves modelling the wind direction on the basis of geographic coordinates, offers an effective method for compensating for such variability and enhancing the accuracy of emission rate estimates.

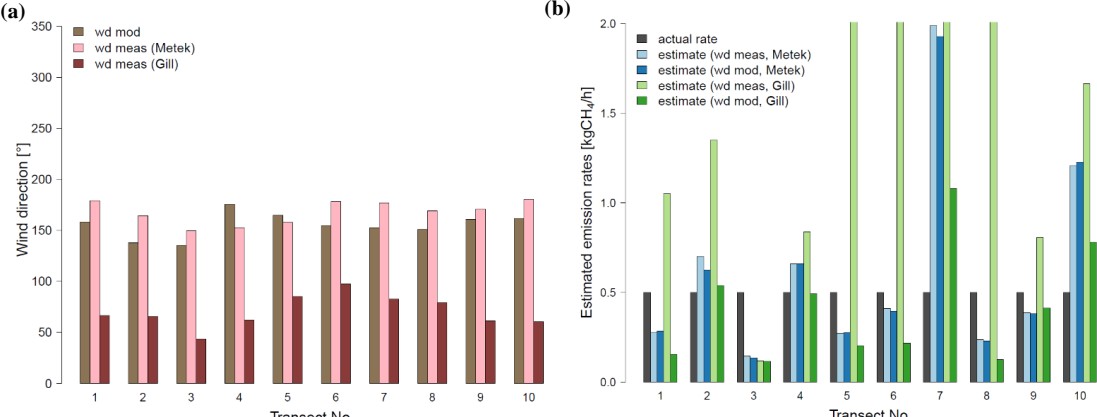

**Figure 6. (a) Measured wind direction for the Metek (green) and Gill (yellow) instruments in comparison with the modelled wind**
**direction shown in blue for 10 transects during the HD1 release experiment. (b) Bar plot showing the corresponding emission rates calculated via the measured and modelled wind directions for the Metek and Gill instruments, respectively, compared with the actual release rates.**

In the study of Korben et al. (2022), the wind speed was averaged separately for each plume crossing, which usually lasted

between 30 and 60 s, whereas in other studies (Riddick et al., 2022a; Kumar et al., 2021; Albertson et al., 2016), it was averaged over longer periods. To determine the influence of the average wind speed, two methods were tested: (1) averaging the wind speed over each individual transect and (2) averaging the wind speed over a set of transects (minimum of 10 transects). In the first case, the transects are considered separately, and the corresponding emission rate for each peak is calculated using the transect wind speed (TWS) that is specifically averaged over this time period. In the second case, the measured wind speed is

usually averaged over a period of approximately 20–30 minutes, and the mean wind speed (MWS) is applied to all peaks and used to calculate the corresponding emission rate. To compare the two methods, a set of transects from the HD1 release experiment is shown in Fig. S11, which shows a bar plot with the actual release rate and the estimated release rates for the calculation via the MWS and TWS using wind measurements obtained with a) the Metek and b) Gill instruments. No significant differences were found between the two methods. Nevertheless, the TWS was chosen to calculate the wind speed

because it allows for a more immediate response to potential wind changes, especially during longer measurement periods, as is often the case during field measurements.

To relate the measured methane concentration to the emission rate, it is essential to consider the stability of the atmospheric boundary layer. In our case, we account for this aspect by categorizing it into a certain PGSC (Section 2.4 and Table S2). This approach was used to describe the dispersion characteristics of the atmosphere and is represented in the Gaussian model by

the dispersion parameters $\sigma_y$ and $\sigma_z$, as described in Section 2.3. The classification impacts the dispersion characteristics of the atmosphere, which are critical in the Gaussian dispersion model. According to Pasquill (1961), the stability of the atmosphere can be described in a simplified way using the wind speed and observations of the position of the sun and cloud cover as a proxy for solar radiation (based on conditions in England). Riddick et al. (2022) used direct measurements of solar radiation to classify high irradiance when the solar radiation exceeds 1000 W m$^{-2}$, moderate irradiance when the solar radiation varies

between 500 and 1000 W m$^{-2}$ and low irradiance when the solar radiation is below 500 W m$^{-2}$. As part of the release

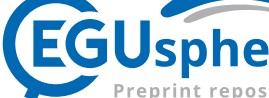

experiments, we were determined how the parameterization of the stability classes affects the calculated emission rates. The solar radiation recorded during mobile surveys on the roof of the Institute of Environmental Physics in Heidelberg (latitude: 49.4172553; longitude: 8.67437285; height a.g.l.: 36 m) was used. The emission rates for each release experiment were calculated using both approaches for categorization into a stability class, and the obtained values were compared with the true

release rates to assess the accuracy of the classification methods, as shown in Fig. 7, which is exemplary for MA1. The emission rates calculated via the original Pasquill classification were affected by the overcast conditions on the experimental day, leading to all releases being assigned to stability class D. This resulted in an average deviation of 36% from the true release rate. In contrast, when the modified approach with the measured solar radiation was used (Riddick et al., 2022), all $CH_4$ releases were assigned to stability class C due to the moderate wind speeds and low solar radiation levels.

The use of the measured solar radiation instead of the cloud cover reduced the average deviation in the calculated emission rates to 8%, resulting in more accurate emission rates. This study highlights the advantages of the direct measurement of solar radiation according to the method of Riddick et al. (2022), providing a more reliable approach to stability classification and minimizing the uncertainty in emission rate calculations. Despite these improvements, the categorization process can still be subject to uncertainties, especially when transitioning between adjacent stability classes (e.g., B–C or C–D).


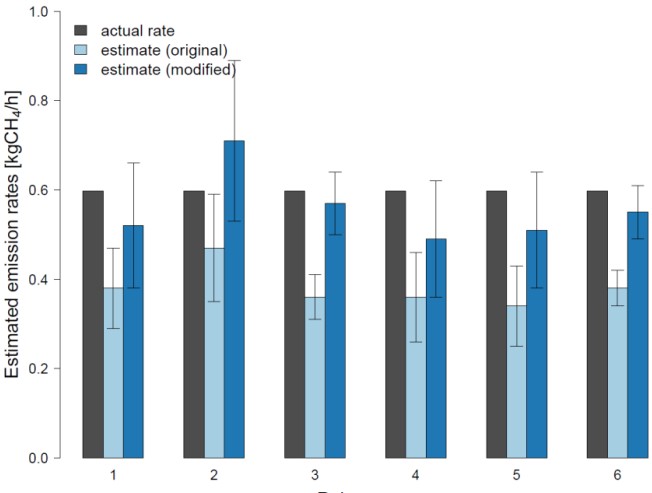

**Figure 7. Bar plot with the actual release rates and estimated release rates (means) for the calculations categorized according to the original classification of Pasquill (1961) and the modified classification of Riddick et al. (2022) for six individual releases during MA1.**

**3.3 Long-term measurements at the biogas plant near Heidelberg**

Between August 2016 and August 2024, 31 mobile measurement campaigns were performed at a biogas plant near Heidelberg, with data originally evaluated with different focuses and methods: 2016–2018 (only the isotopic source signature in Hoheisel et al. (2019)), 2019–2021 (emission rates with different model configurations; Korben (2021)) and recent measurements between 2022 and 2024. As part of this study, all measured data were evaluated using the standardized approach with the

GPM, which had previously been validated through controlled release experiments. Both the driving strategy and the measurement instruments were changed during the eight-year study period. On average, 12 plume crossings were performed per site visit, although the number of crossings ranged from 3 to 36. In total, the methane plume was crossed 372 times on 31 separate days, with 239 (64%) of these transects on 26 measurement days accepted for final analysis. The reasons why




measured transects were not included in the final evaluation were incomplete measurement (turning inside the plume) or the

influence of other emitters, which was determined by the correlation between the model output and the measurement. During these plume crossings, the maximum $CH_4$ mole fractions varied between 2.3 and 51.9 ppm. The GPM, together with the defined evaluation criteria, was used to estimate $CH_4$ emission rates from the recorded methane mole fractions. The daily mean $CH_4$ emission rates, with standard errors of the mean, were calculated from the individual transect emission estimates. These values, shown in Fig. 8, ranged from 0.6 to 13.6 kg $CH_4$ h$^{-1}$ and are documented in greater detail in Table S4.

In addition to change in the trace gas analysers used throughout the measurement process indicated in Fig. 8 by different symbols, the driving strategy was adjusted to ensure that at least 10 transects were completed per measurement. The results of the release experiments revealed a significant reduction in the standard deviation of the emission rates when 10 or more transects were covered. From 2016 to 2019, only 8 transects were covered on average (5 were accepted), whereas from 2020 onwards, an average of 19 transects were traversed (12 accepted), some of them at 2 distances. In addition, from 2024 onwards,

a more accurate stationary wind measurement station was set up during plume crossings. This trend is particularly notable in the error bars of the biogas plant emission rates, where the uncertainty decreased from an average of 40% to 12% after the changes were implemented. This finding is consistent with results from our release experiments (Fig. 3) and the work of Caulton et al. (2018), summarising that 10 transects reliably constrain the influence of atmospheric variability on emission estimates.

The average methane emission rate for the biogas plant was estimated as 5.9 ± 0.5 kg $CH_4$ h$^{-1}$ (52 ± 4 t$CH_4$ yr$^{-1}$). The methane loss rate was defined by Fredenslund et al. (2023) as the site total $CH_4$ emission rate compared with the sum of the $CH_4$ produced and the $CH_4$ emitted into the environment. The calculated loss rate at the Heidelberg biogas plant over the study period ranged from 0.5% to 10.1%, with a mean value of 4.6%. This value occurs within the range reported in similar studies, with Bakkaloglu et al. (2021) providing an estimated average methane loss of 3.7% for biogas plants in the UK, and

Fredenslund et al. (2023) reporting a methane loss of 4.7% at agricultural biogas plants in Denmark. In a recent study by Wechselberger et al. (2025), the average $CH_4$ loss rate at biowaste treatment plants was estimated as 2.8%, which is lower than our estimate.

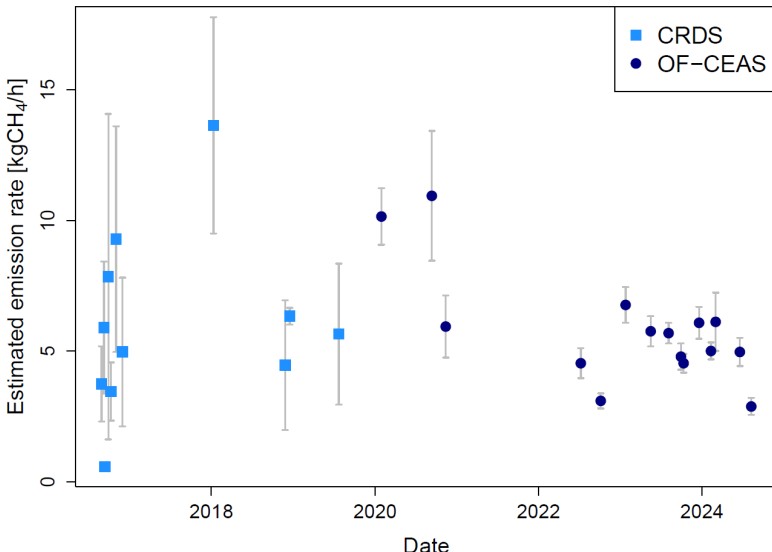

**Figure 8. Harmonized dataset of the estimated mean $CH_4$ emission rates at a biogas plant for each measurement day between 2016**

**and 2024, calculated using the GPM. Emission rate values before 2020 in light blue are based on measurements performed with a CRDS trace gas analyser, and measurements after 2020 were performed via an OF-CEAS instrument with an adapted driving strategy.**



Overall, the calculated emission rates showed greater variability until the end of 2021, with measurements exceeding 10 kg $CH_4$ $h^{-1}$. Starting in 2022, a more consistent mean value emerged ($5.0 \pm 0.5$), with the rates fluctuating only slightly around it. In 2022, the gas storage tank cover was renewed, which could have contributed to the lower and less variable emissions observed thereafter through the use of new low-emission technologies. It is important to note that single measurements only capture emissions at a specific moment, and higher or lower emissions could occur at other times. However, no significant fluctuations were observed during our various measurement campaigns conducted at different times. This is likely because the biogas plant was operating normally, without any flaring or pressure relief. Previous studies have shown that when a plant is not operated optimally, emissions can increase significantly for short periods (Baldé et al., 2022, Wechselberger et al., 2025). Continuous $CH_4$ mole fraction measurements from a network of fixed sensors along with meteorological records could represent an alternative approach to increase the time span, reduce smaller data gaps and provide a dense long-term dataset of methane emissions from biogas plants.

To classify the $CH_4$ emissions of $52 \pm 4$ t $CH_4$ $yr^{-1}$ from the biogas plant in Heidelberg, we can compare them with those of other sources, such as $CH_4$ emissions from natural gas leaks. Wietzel and Schmidt (2023) reported that $CH_4$ emissions from natural gas leaks in Heidelberg (160,000 inhabitants) are of a similar order of magnitude, namely, 42 t $CH_4$ $yr^{-1}$.

**4. Conclusion and recommendation**

Mobile measurements combined with a Gaussian plume model were used to quantify methane emissions from a biogas plant, a method that was investigated in detail during six controlled $CH_4$ release experiments. The findings of these release experiments, in which the true release rates were known, enabled us to improve and standardize the measurement methods and emission rate calculations via the Gaussian plume model. These experiments provided valuable information that contributed to the reliability of mobile sampling as an effective method for emission quantification and showed a measurement uncertainty lower than 30% for the application of the GPM. To better limit the uncertainty in Gaussian emission estimates derived from mobile platforms, we recommend the following improvements:

- **Onsite wind measurements**, ideally supplemented by a stationary anemometer, should be conducted to enhance the accuracy of wind data.
- **Solar radiation measurements** help in determining the atmospheric stability class and improving model predictions.
- **A minimum of 10 transects** should be traversed to account for atmospheric variability and ensure robust results.
- **The use of low driving speeds and high-temporal-resolution instruments**, which are essential for accurate peak measurements, should ideally be employed within the peak area.
- **A sufficient distance** from the emission source, with a recommended minimum value of 20 m, should be maintained to avoid near-field distortion in emission estimates.
- **The measurement and modelling setup** should be verified through controlled release experiments, ensuring a reliable and repeatable method.

The controlled methane release experiments in which the true emission rate was known provided a basis for adapting the measurement method to create a standardized dataset for long-term monitoring. The use of this approach increased the overall accuracy of emission estimates and ensured that variations in methane emission rates were attributable to changes at the biogas plant rather than inconsistencies in measurement procedures or the driving strategy. Crucially, integrating the emission data over the peak area, rather than relying on the peak height alone, minimized the potential impact of analyser changes. In addition, no measurements were performed at wind speeds below 1 m $s^{-1}$, as such conditions were not tested during the release experiments, and measurements under such conditions could therefore lead to further uncertainties.

In this study, we provide a comprehensive dataset of long-term mobile measurements collected over 26 campaigns to quantify methane emissions from a local biogas plant. This dataset provides critical insights into the temporal dynamics of emissions



from such facilities. As methane emissions from biogas plants can vary considerably over time, this variability is often driven

by factors such as unpredictable leakages or release events for safety reasons as well as feedstock type (Wechselberger et al., 2025). Nevertheless, no significant temporal variability in the estimated emission rates was observed in our study. This could be attributed to the fact that the biogas plant operated continuously without major deviations from standard operational procedures during the measurement periods. Such stability in emissions may be less common at other biogas plants, where operational disruptions or maintenance activities could lead to higher or more variable emissions. To better understand methane

emissions and capture a wider range of emission scenarios, it is recommended that measurements be conducted more frequently and over longer periods. Furthermore, ensuring that measurements are conducted under similar meteorological conditions could help standardize emission rate classifications.

An additional strength of our approach is the successful harmonization of the dataset, which spanned 8 years and involved multiple measurement setups. This consistency across different sampling strategies and periods enhances the reliability of the

dataset, facilitating a more comprehensive assessment of methane emissions from the biogas plant.

**Code and data availability**

All raw data can be provided by the corresponding authors upon request.


**Author contribution**

MS and JW designed the study. A. H, P. K and J. W. performed the measurements. J.W. (re-)analysed all data and prepared the emission estimates, J. W. and M.S. prepared the manuscript with contributions from A.H and P. K

**Competing interests**

The authors declare that they have no conflict of interest.

**Acknowledgements**

We thank Henrik Eckhardt, Marvin Seyfarth, Julian Grossmann, Maren Zeleny, Till Gonser and Sarah Reith for

driving/accompanying during mobile measurement campaigns, Johannes Kammerer for his work on the Gaussian plume model code and help during the first release experiments, Carolina Nelson for the measurement campaign in Stuttgart, and Michael Sabasch for the technical support and Roland Pfisterer for his help during our visits at the biogas plant.

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
