# Peer review of "Best Practices and Uncertainties in CH₄ Emission Quantification: Employing Mobile Measurements and Gaussian Plume Modelling at a Biogas Plant"

_EGUsphere, 2025_

## Author Response (AR1)

Comments Referee 1 (https://doi.org/10.5194/egusphere-2025-1344-RC1):

Referee 1: This study systematically and empirically investigates the quantification of methane emissions from biogas plants using mobile measurement and the Gaussian Plume Model (GPM). Based on 26 field campaigns over eight years and six days of controlled release experiments, the temporal coverage and data continuity achieved are rare among similar studies. The author thoroughly examines the reliability of the method and sources of error, successfully constraining the uncertainty of emission estimates to within 30%, thereby significantly enhancing the consistency and credibility of the data. An optimized measurement strategy is proposed, along with practical recommendations for emission monitoring. Overall, the manuscript is well-structured with extensive dataset and good methodology, demonstrating the publication potential on AMT journal. I suggest the publication after addressing the following issues.

Response: We would like to thank the reviewer for the thoughtful and positive evaluation of our manuscript. We are pleased that our comprehensive data set, the robust analysis of our quantification method and our practical recommendations for emissions were well received.. We will carefully consider and address the specific issues raised in order to further enhance the quality and clarity of our work.

Referee 1: In Section 2.4, the extent of simplifications in the Gaussian Plume Model assumptions is not sufficiently discussed. It is recommended to elaborate on the systematic errors induced by neglecting factors such as near-surface turbulence and variations in surface roughness.

Response: We agree that the simplifications of the Gaussian Plume Model should be addressed more explicitly. In the revised manuscript, we have added a sentence acknowledging that the GPM's underlying assumptions, such as neglecting near-surface turbulence and surface roughness variations and simplifying meteorological conditions, can introduce systematic uncertainties. We have also included references (e.g. Hanna et al., 1982, Wilson et al., 1976 and Abdel-Rahman, 2008) to highlight the model's well-documented limitations. The new text emphasizes the trade-off between computational efficiency and the physical accuracy of dispersion predictions. We rewrote Line 173-177: *"The GPM is a relatively simple model that assumes steady weather conditions and a clearly defined plume originating from a single source. Due to its theoretical basis and reliance on idealized assumptions, the GPM neglects several important processes. Notably, it does not account for the effects of near-surface turbulence, temporal and spatial variability in wind fields, or heterogeneity in surface roughness. All of these factors can significantly influence plume behaviour and can introduce systematic errors in emission rate estimates. The empirical relationships used to parameterise dispersion are based on limited datasets and specific experimental conditions, such as those used to parameterize the PGSC (Haugen, 1959), and these relationships may not be applicable to diverse field scenarios. Consequently, the model has been documented as being limited by factors such as averaging time, source distance, atmospheric stability and terrain complexity (e.g. Hanna et al., 1982, Wilson et al., 1976 and Abdel-Rahman, 2008). Despite these limitations, the GPM remains useful in many applications when used with a well-tested setup and an adapted sampling strategy. Its main strengths lie in its low computational requirements allowing efficient modelling of numerous mobile measurements without the need for direct*

*access to the site, making it cost-effective and ideal for achieving high sampling efficiency at individual sources."*

Referee 1: In Section 3.1, it is unclear whether the impact of background gas concentration trends on measurement results during the controlled release experiments has been considered.

Response: We agree that accounting for background concentration trends is critical, especially during controlled release experiments. As noted in Lines 154–157 of the manuscript, we addressed this by applying a variable background fit method (Ruckstuhl et al., 2001; Wietzel and Schmidt, 2023), which accounts for temporal variations in background methane mole fractions. This method ensures that only the $CH_4$ excess attributed to the point source is modelled by the GPM.

Referee 1: In the section "Number of transects and data averaging" (Line 245), mean values under different emission conditions may be influenced by extreme outliers. Has robust statistical testing or alternative statistical approaches been considered to support the selected averaging method?

Response: We agree that the choice of averaging method can affect the final emission estimate, especially when there are outliers present. To address this issue, we compared the performance of mean- and median-based averaging across all six controlled release experiments, rather than just one. Despite its sensitivity to outliers, our analysis showed for the cases we tested that the mean value represents the actual release value better than the median in nearly all release cases when using our best-practice setup. This is also shown in Figures 3 and 4. Based on this empirical evidence, we chose to apply the mean to average individual transect results. For clarification we added in Line 269: *"The median is usually more robust against outliers, but as can be seen in Figure 3 (…)"* and in Line 273: *"As in the previous example, the mean provides a more accurate estimate than the median for almost all of the analysed release experiment results, with an average difference of 35% compared to 43%. Based on this, we have chosen to use the arithmetic mean as the averaging method for our analysis of the measurements performed with our setup."*

Referee 1: The method of error propagation (Line 365) is not fully explained. It is recommended to explicitly describe how errors from wind speed and stability class classifications are propagated into the final emission estimate uncertainty.

Response: For the record, we would like to clarify that we did not perform a formal statistical error propagation of individual input parameters (such as wind speed or stability class) into the final emission estimate. Instead, we assessed the total uncertainty of our approach empirically through controlled release experiments. In the case of the stability class assignment (Line 365), but also for other parameters like number of transects or wind speed averaging strategy, we used the controlled release experiments to show that using the solar radiation we improved *"(…) the average deviation of 36% from the true release rate. (…) The use of the measured solar radiation instead of the cloud cover reduced the average deviation in the calculated emission rates to 8%, resulting in more accurate emission rates."* This empirical uncertainty value accounts for the combined influence of key input parameters under realistic field conditions.

Referee 1: The analysis of long-term emission trends in Section 3.3 is insufficient. Only the emission variation range is described, lacking statistical tests based on time series analysis to quantitatively confirm the existence of significant trends. Further analysis is recommended.

Response: Thank you for this valuable suggestion. We agree that a more quantitative assessment of long-term emission trends would strengthen the analysis. However, even with eight years of measurements, the data is intermittent and the number of data points is limited. This means that we do not have the statistical confidence to identify trends that can be attributed solely to plant emissions. As discussed in Lines 395–404, any observable trends are more likely to be due to changes in the measurement strategy over time. Further some fluctuations may also reflect operational changes, as in 2022, the cover of the gas storage facility was renewed (Line 420), which may have contributed to the lower and more stable emissions observed since then through the use of new low-emission technologies.

Referee 1: There is a lack of discussion regarding exceptionally high emission events. For instance, Table S4 in the Supplementary Material shows early instances of single-event $CH_4$ emission rates exceeding 10 kg/h. It is suggested to provide a supplementary analysis of possible causes (e.g., process failures or maintenance activities).

Response: Thank you for this valuable comment. We agree that exceptionally high emission events warrant further discussion. As noted, some of the early measurements (e.g., values exceeding 10 kg/h in Table S4) indicate elevated emission rates. There are several possible explanations for these outliers. First, during the initial campaigns, our driving and sampling strategy was still under development, which may have contributed to greater variability or overestimation in individual measurements. Second, we cannot exclude the possibility of temporary leaks or operational anomalies during those periods, although we did not receive any reports from the biogas plant operator indicating known failures, venting, or maintenance activities on the respective days. However, noticeably high measured $CH_4$ concentrations on the three days with particularly high emission rates (> 10 kg/h) indicate an irregular methane release. To take this into account, we have added a sentence in Line 419 with according references: *"However, the exceptionally high measured concentration of up to 52 ppm $CH_4$ on the three days of particularly high emission rates (> 10 kg $CH_4$ $h^{-1}$) suggest irregular methane release, possibly due to pressure relief or leakage. Previous studies have shown that emissions can increase significantly for short periods when a plant is not operated optimally (Baldé et al., 2022, Wechselberger et al., 2025)."*

Referee 1: The comparison with existing studies (Line 410) is rather superficial. A systematic comparison table between this study and datasets such as Fredenslund et al. (2023) and Bakkaloglu et al. (2021) is recommended, along with a deeper analysis of the underlying causes of observed differences (e.g., facility scale, measurement methods).

Response: We agree that the comparison with existing studies in the original manuscript was relatively brief and could benefit from a more systematic and in-depth analysis. In response, we have added a new table (Table S5) summarizing key characteristics of our study and of relevant datasets from Bakkaloglu et al. (2021), Fredenslund et al. (2023), and Wechselberger et al. (2025). The table includes information on $CH_4$ loss rates, measurement methods and plant types to support a structured comparison.

Additionally, we have expanded the discussion in Section 3.3 to consider the reasons for the differences observed in methane loss rates. This provides more context for interpreting our results in relation to the wider literature and strengthens the overall interpretation of our findings. We added to Line 412: *"The variation in reported methane loss rates among these studies is due to differences in several key factors such as different measurement techniques that influence spatial and temporal resolution, feedstock composition, plant type and technology (e.g. gas tightness, storage cover and digestate handling), can also influence CH₄ losses and which can vary significantly between sites and countries. Despite these methodological and contextual differences, the CH₄ loss rate observed in this study is broadly consistent with values reported in the literature, highlighting the importance of our approach when combined with an adapted sampling strategy."*

Referee 1: The model applicability under ultra-low wind speed conditions (<1 m/s) is mentioned (Line 455) without adequate explanation. Although low wind speed conditions were not tested, why is the model deemed unsuitable? Could this lead to underestimation of emissions?

Response: We agree that the applicability of the model in conditions of ultra-low wind speed (<1 m/s) requires further clarification. Although our release experiments did not include such conditions, our assessment is based on several years of field experience using the Gaussian plume model, as well as supporting literature. (e.g. Wilson et al., 1976; Essa et al., 2005; Mortarini et al., 2016). We have clarified this in the revised manuscript in Line 456 and added references to support this point: *"Based on both field experience and existing literature (e.g., Wilson et al., 1976; Essa et al., 2005; Mortarini et al., 2016), ultra-low wind speeds present significant challenges for Gaussian plume modelling. Under these conditions, enhanced plume meandering and atmospheric instability lead to increased lateral and vertical dispersion, which the model does not adequately account for. As a result, the model shows a tendency to underestimate emissions. For this reason, data collected at wind speeds below 1 m s⁻¹ are associated with high uncertainty and were excluded from the analysis."*

Referee 1: Line 465 mentions the impact of facility operational anomalies on emissions. During the eight-year data collection period, was there any inquiry into the biogas plants regarding maintenance, accidents, or special operational conditions? How might these events have influenced emission stability?

Response: Thank you for this insightful comment. During the eight-year monitoring period, we maintained regular contact with the plant operator and were informed of major changes to the installation. Although some structural modifications were made during the eight years, most notably the renewal of the gas storage tank cover in 2022 (as mentioned in Lines 424-428), we were informed that the plant operated continuously at a stable power level with no significant variations in gas production or processing capacity. To our knowledge, there were no venting events, maintenance activities, or operational issues on the specific days when measurements were conducted. This operational stability supports the interpretation that the emission patterns observed were representative of normal plant functioning, and not significantly influenced by short-term variations. (see Line 466-468).

Referee 1: In practical biogas plant areas, multiple emission sources may exist. How was single-source contribution confirmed and source mixing errors avoided? How to ensure that current methods are reliable?

Response: In order to identify and localize the dominant emission points at the biogas plant and confirm the primary emission source, we conducted walking surveys on-site to avoid errors due to source mixing. These verification measurements were carried out several times between 2022 and 2024, but were not carried out during earlier field campaigns. This could introduce an error if the methane was mainly emitted from a different source. Combining on-foot source identification with spatially flexible mobile measurements significantly improves the reliability of our method in multi-source environments. Therefore we add to Line 404: *"Further, the location of the emission source was determined by walking surveys and identifying the area with the highest methane mole fraction. These verification measurements were carried out several times between 2022 and 2024, but were not carried out during earlier field campaigns. This could introduce an error if the methane was mainly emitted from a different location on the plant. Combining on-foot source identification with spatially flexible mobile measurements significantly improves the reliability of our method."*

Referee 1: Although an area-averaging method was employed to reduce bias, was the potential long-term drift between CRDS and OF-CEAS instruments assessed, which could lead to systematic errors?

Response: To minimize potential long-term drift and systematic errors between the CRDS and OF-CEAS instruments, regular measurements with certified calibration gases were performed throughout the time. Calibration procedures followed the protocols outlined by Hoheisel et al. (2019) for the CRDS and by Korben et al. (2022) and Wietzel and Schmidt (2023) for the OF-CEAS analyser (see Lines 122 and 130 of the manuscript). These regular checks ensured the consistency and comparability of concentration measurements over time and between instruments.

Referee 1: Issues related to figure quality:

• In Figure 2 (Line 180), the sizes and y-axis font sizes of panels (a) and (b) are inconsistent. It is recommended to adjust them and check other figures for similar issues.

• In Figure 3 (Line 265), the border styles between panels (a) and (b) are not uniform, and the boxplot color distinctions are not sufficiently clear. Revisions are advised.

• In Figure 6 (Line 335), the legend "Metek (green) and Gill (yellow)" does not correspond to the actual colors in panel (a). It is recommended to correct the legend and adjust the color scheme for better contrast.

• Some figures, such as Figures 6 and 7, lack borders, while others include them. It is recommended to standardize figure formatting throughout the manuscript.

• In Figure 8, why is there no error bar for one CRDS-based $CH_4$ emission estimate? Was the

error too small to be visible, or was it omitted? Please clarify.

Response: Thank you for your detailed comments regarding the figures. We have carefully reviewed all figures and made the following improvements:

Standardized the font sizes and panel dimensions in Figures 2 and 3 to ensure consistency.

Unified border styles and enhanced colour distinctions in Figures 6, and 7 for clarity and better visual contrast.

Corrected the legend colours in Figure 6(a) to accurately reflect the data.

Added borders where needed to standardize figure formatting throughout the manuscript.

Clarified in the Figure 8 caption that the error bar for the CRDS-based $CH_4$ emission estimate is present but very small, making it difficult to see.

Comments Referee 2 (https://doi.org/10.5194/egusphere-2025-1344-RC2):

Referee 2: Review of "Best Practices and Uncertainties in CH4 Emission Quantification: Employing Mobile Measurements and Gaussian Plume Modelling at a Biogas Plant" by Wietzel et al., for publication in AMT

This manuscript describes how the quantification method based on mobile measurements combined with a Gaussian plume model to assess methane emissions from local sources can be improved and give practical recommendations. The strength of the manuscript is the extended study, which incorporates a total of six controlled release experiments, carried out over several years. Various parameters and driving strategies were tested such as the influence of the distance from the source, the number of transects, the averaging strategy or the influence of meteorological parameters, to accurately estimate methane emissions when using mobile measurements combined with a Gaussian plume model. These methodological improvements were integrated into the long-term (8 years) emission monitoring of a biogas plant in Heidelberg, Germany and helped to reduce the uncertainty of the estimated methane emissions for this site.

The manuscript is well-written and logically structured. It addresses scientific questions relevant to the scope of AMT. For the reasons mentioned above, the paper is appropriate for publication in AMT after few minor revisions described below.

Response: We thank the reviewer for the positive and encouraging feedback. We appreciate the recognition of the of the strengths of our study, including the extensive controlled release experiments, the methodological improvements and the long-term application of the approach. The suggested minor revisions will be addressed carefully in the revised manuscript to further improve the quality of our work.

Referee 2: Table 1: This is a nice table, it could be useful to add other information such as the standard deviation of the measured wind speeds during the different controlled release experiments to have a better idea of the stability of the weather conditions during each experiment. Rates emitted during the different release steps could also be interesting (or at least a range).

Response: Thank you for the valuable suggestion. To provide a clearer assessment of the stability of the weather conditions and release parameters during each controlled release experiment, we will include the ranges (minimum–maximum) of measured temperature, wind speed and adjusted release rates in Table 1.

Referee 2: L. 212-213: "Similar to Takriti et al. (2021), we found that for our two analysers, the peak heights differed, but the area under the molar fraction curve integrated over time remained consistent." It would have been nice to show in the Supplementary Material the two plots: peak height CRDS vs. peak height OF-CEAS and peak area CRDS vs. peak area OF-CEAS (just like in Takriti et al. (2021)) based on your data to prove your point.

Response: Unfortunately, in our study, the temporal overlap between the two analyzers during mobile operation was only during one measurement day, resulting in an insufficient number of peaks for a meaningful comparison plot as shown in Takriti et al. (2021). The approach using the peak area and not the height has also been adopted in other studies (e.g.

Mønster et al., 2014; Korben et al., 2022). We will clarify in the revised manuscript with the following sentence in Line 207: *"In our study, both measuring devices were used in parallel for mobile measurements on just one day. During this field campaign at a wastewater treatment plant, we investigated the response time of the CRDS and OF-CEAS analysers, which were operated simultaneously with different temporal resolutions."* And further Line 213: *"(…) thereby confirming the findings of Takriti et al. (2021)."*

Referee 2: L. 219: "No dependencies of the determined emission rate on these two modifications were found". This sentence suggests that neither driving speed nor type of mobile vehicle impact the emission estimates, but it is recommended to use low driving speeds for accurate peak measurements in the conclusion. This part should be changed a bit to explain why the authors recommend to use low driving speeds in the conclusion.

Response: We have revised the relevant section to clarify while no significant impact of driving speed or vehicle type on the emission estimates was observed during the controlled release experiments, driving speeds of around 30 km h$^{-1}$ still has advantages. Slower speeds allow for a denser sampling of the $CH_4$ concentration peaks, leading to improved temporal resolution. This clarification has been added to Line 218 in the manuscript: *"No significant dependence of the determined emission rate on the choice of vehicle or driving speeds between 20 km h$^{-1}$ and 50 km h$^{-1}$ was found. However, to achieve higher temporal resolution of the $CH_4$ peaks, a low driving speed of around 30 km h$^{-1}$ is recommended, as it enables denser sampling of the $CH_4$ mole fraction signal. This is particularly important when sampling close to the source, where plume peaks tend to be sharper and less Gaussian in shape."*

Referee 2: L. 242-243: "In this respect, with our measurement and modelling setup, we would only accept measurements obtained at a distance greater than 20 m from the emitter." The authors should also mention that this distance criterion only works for sites where the location of the sources is precisely known (as explained in the previous sentence).

Response: We acknowledge that uncertainties in the source location can influence emission estimates, particularly at short distances. However, our main point of this section is that, based on our controlled release experiments, our measurement and modelling setup does not provide reliable emission estimates at distances below 20 m from the source, even when the source location is known accurately. While source location uncertainty is an additional factor, the >20 m criterion reflects a methodological limitation of our approach, as supported by the experimental results (see MA3 and HD2). To improve clarity, we have added a sentence: *"This criterion applies regardless of how accurate the source location is known, but the uncertainty from an imprecise source position needs to be considered even at distances greater than 20 m, and the impact of this uncertainty decreases as the distance from the source increases."*

Referee 2: L. 281: Problem with the title numbering, should be 3.2.

Response: Thank you for noticing this. The section numbering has been corrected in the revised manuscript.

Referee 2: Fig 6 (a): wrong colors in the legend.

Response: We have corrected the legend of Figure 6(a).

Referee 2: L. 346: Has the wind averaging strategy comparison TWS/MWS only been applied to transects from the HD1 controlled release experiment? It is not really clear in the manuscript as only results from this experiment are shown in the supplementary material. It would be interesting to test this strategy over the entire controlled release dataset to draw more robust conclusions and check if one of the approaches could be more suitable for specific weather conditions (wind speed, wind variability, etc…).

Response: A comparison of the TWS and MWS wind averaging strategies was conducted across the full set of controlled release experiments. The results were very similar in all cases, with no significant differences in model performance attributable to the choice of wind averaging method. To improve clarity, only HD1 is shown in the supplementary material, as it is representative of the overall findings. Nevertheless, we have prepared an additional figure for the supplementary material to illustrate the effect of different weather conditions.

[Figure]

*"Figure S12: Comparison of $CH_4$ emission rates calculated using transect wind speed (TWS) versus those calculated using mean wind speed (MWS) for each controlled release during HD1–HD3. Data points are color-coded according to the measured wind speed during each release. The dashed line represents the 1:1 reference line."*

Referee 2: L. 365: Same question for the parameterization of the stability class: why only show results from MA1 instead of all the controlled release experiments together?

Response: Here, we chose to present the results from experiment MA1 as a representative example to illustrate the potential influence of stability class selection on the output of the GPM. In this case, the impact is particularly pronounced due to the overcast conditions during the release. Following standard classification schemes, this would have led to an assignment to stability class D, as discussed in Lines 370–373. By selecting this example, we aimed to demonstrate how such a seemingly straightforward classification can significantly affect model performance. Including all controlled release experiments would have extended the discussion considerably without necessarily adding further clarity to the key point that careful consideration of stability class assignment is critical, especially under ambiguous meteorological conditions.

Referee 2: L. 361: "we determined" rather than "we were determined".

Response: The sentence has been corrected accordingly.

Referee 2: L. 444-445: This conclusion about using low driving speeds should be a bit nuanced in my opinion. I understand the explanation given in the Supplementary Material about the better resolution of the peak (8 points at 20 km/h versus 5 points at 50 km/h). But these plumes were collected at 25 m of the source which is pretty close. At this distance, peaks are usually narrower/spikier than when transects are measured further away from the source and tend to look more Gaussian. So, I agree that for spiky peaks collected close to the source, missing data points near the maximum height of the plume can have a large influence on the measured area and estimated emissions, however I would argue that it is not as important when transects are measured further away and plumes look Gaussian.

Response: Thank you very much for this valuable comment. We agree that the shape of the plume changes with distance from the source, becoming smoother and more Gaussian-like further downwind. This affects the importance of data resolution. Our conclusion regarding lower driving speeds was primarily based on experiments conducted at a short distance (25-50 m), where the plume tends to be more concentrated and spiky. Here, undersampling can lead to a noticeable underestimation of peak concentrations, and thus of emission estimates. We appreciate your suggestion to nuance this conclusion, and we will revise the manuscript accordingly to clarify that the benefit of lower driving speeds is particularly relevant for transects close to the source where the structure of the plume is highly variable over short distances. For measurements taken farther away, where the plume is broader and smoother, the impact of sampling resolution on the total integrated concentration may be less pronounced. We believe that this clarification will strengthen the practical guidance for mobile measurement strategies under varying conditions.

To reflect this important suggestion, we have made the following changes:

- We added the sentence *"It should be noted, however, that this effect is particularly relevant for measurements taken close to the source, where plumes are narrower and more variable; at greater distances, where the plume shape tends to smooth out and become more Gaussian, the influence of driving speed on emission estimates may be less significant."* to the Supplementary Material.
- Additionally, the phrase *"especially close to the source."* has been added to line 445 to further emphasize the context of our recommendation.

These adjustments aim to clarify the applicability of the driving speed recommendation and acknowledge the variability in plume structure depending on distance from the source.

Referee 2: L. 446: The recommendation of performing plume measurements at a minimum distance of 20 m from the source should also be nuanced and it should be clearly stated that measurements at such a short distance from the source only works when the sources' location is precisely known.

Response: Thank you for your comment. We acknowledge that the >20 m minimum distance recommendation reflects both a limitation specific to our measurement and modelling setup and a general limitation of the Gaussian plume model, which tends to produce unrealistic (infinite) values very close to the source. While some studies have reported reliable results at shorter distances (e.g., 15 m), our controlled release experiments consistently showed that reliable emission estimates require measurements beyond 20 m in our configuration. This

highlights the importance of verifying any measurement and modelling approach through controlled release experiments to ensure accuracy and repeatability. Accordingly, we have revised the manuscript to clarify: *"A sufficient distance from the emission source, with a recommended minimum value of 20 m, should be maintained to avoid near-field distortion in emission estimates caused by model limitations near the source and the limitations of our setup."*